# Non-Coding RNAs in Glioma

**DOI:** 10.3390/cancers11010017

**Published:** 2018-12-22

**Authors:** Ryte Rynkeviciene, Julija Simiene, Egle Strainiene, Vaidotas Stankevicius, Jurgita Usinskiene, Edita Miseikyte Kaubriene, Ingrida Meskinyte, Jonas Cicenas, Kestutis Suziedelis

**Affiliations:** 1Nacional Cancer Institute, Santariskiu str. 1, LT-08660 Vilnius, Lithuania; ryte.rynkeviciene@nvi.lt (R.R.); julija.fadejeva@nvi.lt (J.S.); egle.strainiene@nvi.lt (E.S.); vaidotas.stankevicius@nvi.lt (V.S.); jurgita.usinskiene@nvi.lt (J.U.); edita.kaubriene@nvi.lt (E.M.K.); 2Institute of Biosciences, Life Sciences Center, Vilnius University, Sauletekio ave. 7, LT-08412 Vilnius, Lithuania; 3Department of Chemistry and Bioengineering, Vilnius Gediminas Technical University, Sauletekio ave. 11, LT-10122 Vilnius, Lithuania; 4Institute of Biotechnology, Vilnius University, LT-10257 Vilnius, Lithuania; 5Faculty of Medicine, Vilnius University, M.K. Cˇiurlionio 21, LT-03101 Vilnius, Lithuania; 6Proteomics Center, Institute of Biochemistry, Vilnius University Life Sciences Center, Sauletekio al. 7, LT-10257 Vilnius, Lithuania; ingrida.meskinyte@gmail.com (I.M.); j.cicenas@mapkinases.eu (J.C.); 7MAP Kinase Resource, Bioinformatics, Melchiorstrasse 9, 3027 Bern, Switzerland; 8Energy and Biotechnology Engineering Institute, Aleksandro Stulginskio University, Studentų g. 11, LT-53361 Akademija, Lithuania

**Keywords:** non-coding RNA, glioma, miRNA, lncRNA

## Abstract

Glioma is the most aggressive brain tumor of the central nervous system. The ability of glioma cells to migrate, rapidly diffuse and invade normal adjacent tissue, their sustained proliferation, and heterogeneity contribute to an overall survival of approximately 15 months for most patients with high grade glioma. Numerous studies indicate that non-coding RNA species have critical functions across biological processes that regulate glioma initiation and progression. Recently, new data emerged, which shows that the cross-regulation between long non-coding RNAs and small non-coding RNAs contribute to phenotypic diversity of glioblastoma subclasses. In this paper, we review data of long non-coding RNA expression, which was evaluated in human glioma tissue samples during a five-year period. Thus, this review summarizes the following: (I) the role of non-coding RNAs in glioblastoma pathogenesis, (II) the potential application of non-coding RNA species in glioma-grading, (III) crosstalk between lncRNAs and miRNAs (IV) future perspectives of non-coding RNAs as biomarkers for glioma.

## 1. Introduction

Glioma is one of the most aggressive and common primary tumors (~30%) of the central nervous system (CNS), which includes astrocytoma, oligodendroglioma, ependymoma, medulloblastoma and glioblastoma. Despite the progress in surgical resection, radiotherapy and chemotherapy technologies, the prognosis for glioma patients remains poor [1]. The median survival for patients with a high-grade glioma is only 15 months [2]. A better understanding of tumor characteristics and biology is necessary for better tumor classification and patient stratification. Over the past decades, the bulk of glioma and other cancer research was pointed towards the identification of altered gene expression, mutations and epigenetic modifications, which could serve as a molecular biomarkers [3]. In 2016 a new and improved classification of gliomas was approved as a direct result of this research [4]. This new classification takes molecular biomarkers into account thus allowing for more precision. Diffuse astrocytic and oligodendroglial tumors (WHO II and III) are now separated according to IDH mutation and 1p/19q-codeletion status. Also grade IV gliomas are now divided into IDH wild type, IDH-mutant, glioblastoma not otherwise specified (NOS) and diffuse midline glioma, H3 K27M-mutant. Of all adult brain tumors, about 32% are malignant. 46.88% of malignant tumors are glioblastomas. Another large group of brain tumors are diffuse astrocytomas–17.19%. Oligodendroglioma and ependymal tumors comprise 4.69% and 6.25% of adult brain tumours, respectfully. The remaining 25% are other malignant tumors [5].

Diffuse low grade gliomas (WHO II) such as diffuse astrocytoma and oligodendroglioma are classified into three subgroups: (a) low-risk low-grade, IDH-mutant +/− 1p/19q co-deleted tumors. Patients are usually less than 40 years old. More than 14 years of survival can be achieved after maximal safe resection combined with radiotherapy; (b) high-risk low grade large (>5 cm) tumors, patient age over 40, IDH-wild type, often treated as glioblastoma. Survival prognosis is about 5 years; (c) IDH-mutant high-risk low grade gliomas, progression-free survival is 3–5 years. Recent phase 3 trial RTOG 9802 showed that initial radiation followed by procarbazine, lomustine and vincristine (PVC) prologs the survival from 4.0 to 10.4 years [6].

Diffuse high-grade gliomas (WHO III) include anaplastic astrocytoma IDH-mutant, IDH-wild type and NOS, anaplastic oligodendroglioma IDH-mutant/1p/19q co-deleted and NOS. CATNON clinical 3 phase trial revealed that anaplastic astrocytoma with intact 1p/19q status with additional 12 cycles of temozolomide to radiotherapy prolong progression-free survival from 1.5 to 3.6 years [7]. The usage of PVC instead of TMZ is investigated in CODEL trial [8].

Glioblastoma (WHO IV) is the most lethal and most common brain cancer, with a very short survival. Only about 5% of patients survive beyond 5 years. The standard treatment for good performance patients consists of radiotherapy to 60 Gy over 3 weeks with daily TMZ, followed by at least six cycles of adjuvant TMZ [9]. Recent trials reported increased survival with the addition of electric field therapy. It was shown that electric field therapy can prolong survival by nearly 5 months [10].

However, there is still a lot of research needed to fully understand the regulation of cell processes in order to make use of new biomarkers and new technologies as tools to improve the treatment of glioma. Non-coding RNAs, which are involved in global regulation of cellular processes, can be used as potential biomarkers in cancer [11].

Whole genome sequencing revealed that more than 90% of the human genome is transcribed and only ~3% of the genome contains protein-coding genes [12]. The rest of the genome encodes mainly non-coding RNA. These RNAs are split into two classes: housekeeping and regulatory RNAs. tRNA, rRNA, snRNA and snoRNA belong to these housekeeping RNA classes responsible for maintaining a constant protein expression level in the cells. The rest of non-coding RNAs regulate gene expression and are divided into two subclasses according to their length. RNAs that are less than 200 nt represent the subclass of small non-coding RNAs (sncRNA). In addition, sncRNAs include a group of related RNAs such as miRNA, piRNA and siRNA. If non-coding RNAs are longer than 200 nt, they are classified as long non-coding RNAs (lncRNAs) [13].

LncRNAs are mainly polyadenylated 200 nt—100 kb long transcripts, usually transcribed by RNA polymerase II and controlled by multiple type of transcriptional factors. LncRNAs is the largest (>80%) and a very heterogeneous group of ncRNAs, but is less conserved than miRNA [14]. and According to their position in the genome, all lncRNAs can be divided into the following subclasses: sense, antisense, bidirectional, intronic and intergenic [15]. Transcription of lncRNAs may affect the down-stream gene expression either positively or negatively by directly interfering with promoters or by modifying the structure of chromatin. LncRNAs also bind specific proteins such as PcG, Enhancer of zeste homolog 2 (Ezh2), a histone methyltransferase and Polycomb-repressive complex 2 (PRC2) by modulating protein activity, altering its localization or changing its structural or organizational role [16]. Thus, lncRNA-protein interactions may modulate the activity of other protein-binding partners. Recent genome studies suggest that long non-coding RNAs serve asprecursors for small RNAs [17]. Several reports prove that lncRNAs cause mRNRs to be processed into small RNAs [18]. Recent studies indicate that interactions between miRNAs and lncRNAs are able to regulate each other’s expression, thus forming a complex regulatory network which plays an important role in the cell pathophysiological processes. A cross-regulation between lncRNAs and miRNAs includes processes such as miRNA triggered lncRNA decay, lncRNA affected miRNA levels and function regulation, lncRNA competition with miRNA for interaction with mRNAs and miRNA production from lncRNAs [19,20]. The aberrant expression and cross-regulation of miRNAs and lncRNAs can be used as potential diagnostic and therapeutic tools in different malignancies, including gliomas [21,22,23,24,25,26,27,28,29]. Indeed, this confirms that miRNAs, lncRNAs and other non-coding RNAs regulate a number of target genes and play an important role in glioma carcinogenesis, thus serving as predictive and prognostic biomarkers of glioma.

LncRNAs are involved in biological processes such as cell death, growth, differentiation, epigenetic regulation, genomic imprinting, alternative splicing, and regulation of gene expression at the posttranscriptional level, chromatin modification, inflammatory pathologies and subcellular transport. Invalid lncRNAs expression can be associated with cancer development, progression and metastasis formation and can be used as a therapeutic target in glioma [30,31].

Here we provide an overview of the latest studies related to lncRNAs, miRNAs, which directly target reviewed lncRNAs and other small noncoding RNAs that exhibit altered expression in glioma tissue, with a focus on their biological and therapeutic roles in brain tumors and discuss future perspectives.

## 2. Expression of Non-Coding RNAs in Cell Proliferation, Migration, Invasion and Apoptosis

### 2.1. Up-Regulated lncRNAs in Biological Glioma Processes

Cell proliferation, invasion and migration are the major characteristics of cancer cells. The ability of cancer cells to migrate and invade into the normal tissue plays a critical role in patient survival [32]. Specialized cancer cell interactions with the extracellular matrix (ECM) and adjacent cells, accompanied by various biochemical processes support active cell movement [33]. These cells release a number of growth factors such as TGF-β (Transforming Growth Factor-beta), VEGF (Vascular Endothelial Growth Factor), PDGF (Platelet-Derived Growth Factor), FGF-2 (Fibroblast Growth Factor), alter the levels of transcription factors (Twist, Snail, ZEB) required for the initiation of the epithelial mesenchymal transition (EMT) and also release numerous proteases that promote cancer cell invasion into normal brain tissue [34]. Moreover, glioblastomas display a deregulated apoptotic pathway with high levels of anti-apoptotic family proteins Bcl-2 (B-cell lymphoma 2) and PI3K (Phosphatidylinositol-4,5-bisphosphate 3-kinase) [35]. The cell cycle control mechanisms regulated by p53 and RB (Retinoblastoma) proteins are inactive and allow unregulated cell cycle progression and tumor growth [36]. Malignant cell growth is characterized by loss of cellular identity, increased proliferation abilities and deregulation of cell death. Study results indicate that non-coding RNAs are important in this cancer-relevant cellular phenotype regulation [37].

The growing list of long-non-coding RNAs, the expression of which is up-regulated in glioma, supports their importance in proliferation and malignancy processes (Figure 1 and Table 1).

#### 2.1.1. AB073614 lncRNA

The up-regulation of lncRNA AB073614 is associated with a poor survival in glioblastoma patients. A suppressed expression of AB073614 reduces cell migration by increasing *e-cadherin* and decreasing *vimentin* expression. All this data shows the importance of AB073614 in EMT process [38,40]. In 2017, Wang revealed that AB073614 alters proliferation via the PI3K/Akt pathway and is able to increase the expression *MMP9* (Matrix Metallopeptidase 9), *Bcl-2* while decrease the expression of *Bax* [39].

#### 2.1.2. ATB, H19, ZEB1-AS lncRNAs

Lnc-RNA ZEB1-AS1 regulates the expression of *cyclin D*, *CDK2* (Cyclin Dependent Kinase 2), *ZEB1* (Zinc Finger E-Box Binding Homeobox 1), *MMP2*, *MMP9*, *N-cadherin* and *integrin-β1* genes. This proves that ZEB1-AS1 regulates the EMT processes and is involved in proliferation, apoptosis and metastasis of glioma, but data about the exact signaling pathway is lacking [117]. Zhao et al. also showed that elevated H19 expression modulated glioma growth by targeting *iASPP* via miR-140 [60]. Zhang et al. demonstrated that H19 role in proliferation is mediated by miR-675, which is encoded in H19 1 exon [61] and directly suppress Cyclin Dependent Kinase 6 (*CDK6*) expression [62]. An increased expression of lncRNA ATB positively correlates with glioma grade and negatively with the survival rate of glioma patients. It was shown that lncRNA ATB is directly targeted by miR-200a, which affects the expression of *TGF-β2*. TGF-β2, in turn, activates the MAPK signaling pathway [119]. In addition, the down-regulation of ATB in vivo suppressed tumor growth [42]. MiR-152 is another miRNA down-regulated by H19 [120]. This leads to the activation of tumor proliferation and invasion in vitro and in vivo [42].

#### 2.1.3. CCAT1, CCAT2, CCND2-AS1 lncRNAs

LncRNA CCAT1 is involved in the cell proliferation process and is a target of miR-410 [47]. The expression of miR-410 is suppressed in glioma tissue, and this leads to the activation of the MET/Akt signaling pathway [121]. However, there is no evidence that CCAT1 is involved in the regulation of *met* expression. Later, in 2017, another group announced that CCAT1 enhances Fibroblast Growth Factor Receptor 3 (*FGFR3*) and Platelet-Derived Growth Factor Receptor A (*PDGFRA*) expression by targeting miR-181b. Both growth factors activate MAPK/ERK1 signaling pathways [122]. The ability of CCAT1 to promote tumor growth was also shown in vivo [48]. An increased expression of CCAT2 and CCND2-AS1 induces proliferation, cell cycle progression and migration via the Wnt/β-catenin signaling pathway and downregulation of CCAT2 expression significantly reduces brain tumor growth in vivo [49,51].

#### 2.1.4. CRNDE lncRNA

The lncRNA CRNDE is highly up-regulated in glioma tissue, and promotes proliferation, migration and invasion processes [52]. The capability of CRNDE to induce gliomagenesis has been tested in vivo, and it was shown that CRNDE acts via the mTOR signaling pathway, involving *myc*, *cycline-D1*, *p53* and Phosphatase and Tensin homolog (*PTEN*) genes [54]. CRNDE is also a target of miR-384, and both of them regulate the level of PIWIL4 protein. PIWIL4, in turn, promotes progression by inducing phosphorylation of STAT3, the main member in PI3K/Akt1/IL-6/STAT3 pathway [53]. In 2017, it was determined that an up-regulated expression of CRNDE induces EGFR (Epidermal Growth Factor Receptor) activation, resulting in poor survival in glioma patients and apoptosis inhibition through increased Bcl2/Bax (BCL2 Associated X) ratio [55].

#### 2.1.5. CCDC26, FER1L4, miR210HG, MVIH, SPRY4-IT1, TP53TG1, lncRNAs

CCDC26 is a newly identified lncRNA the expression of which is up-regulated in glioma tissue. CCDC26 directly targets miR-203, and this effect was shown in vitro and in vivo [50]. LncRNAs FER1L4, TP53TG1, SPRY4-IT1, miR210HG, MVIH are highly expressed in glioma and their elevated expression is associated with a poor prognosis. They activate the proliferation and invasion processes together with an inhibition of apoptosis via yet undetermined pathways [57,89,90,101,102].

#### 2.1.6. ECONEXIN, TP73-AS1, Xist lncRNAs

An increased expression of lncRNA TP73-AS1 is associated with poor survival in patients, and the down-regulation of TP73-AS1 inhibited the proliferation and invasion processes together with High-Mobility Group Box 1 (*HMGB1*) expression, which plays an important role in many diseases including cancer by targeting miR-142 [103,123,124]. A decreased expression of miR-142 correlates with increased levels of Ras-related C3 botulinum toxin substrate 1 (Rac1), which activates multiple pathways [125]. LncRNA ECONEXIN is up-regulated in glioma tissue and promotes cell proliferation via sponging miR-411-5p and altering the expression of topoisomerase 2 alpha (*TOP2A*) gene [56]. LncRNA Xist also regulates the expression of *Rac1* by inhibiting the expression of miR-137 [112]. Another study provided evidence that direct Xist binding to miR-152 promotes the formation of glioma [114]. Xist also binds miR-29 and miR-429 [115,116]. The fact that Xist binds to many miRNAs proves it’s importance in gliomagenesis, however, the exact pathways are still unclear.

#### 2.1.7. FOXD3-AS1, Linc-OIP5, ZFAS1lncRNAs

In 2017, an lncRNA named ZFAS1 was detected in glioma tissue. ZFAS activates cell proliferation, invasion and migration processes by activating EMT and Notch signaling pathways. Gao et al. also showed that ZFAS1 activates the EMT pathway [118]. There is no data about the interaction of ZFAS1with any miRNAs yet. Linc-OIP5 is another newly identified long non-coding RNA, up-regulated in glioma tissue and positively correlating with a glioma grade. It induces proliferation and migration processes through Notch-1, yes-associated protein 1 (YAP), Jagged-1 (Jag-1) and hairy and enhancer of split-1 (Hes-1) and the down-regulation of its expression reduces tumor growth in vivo [78]. LncRNA FOXD3-AS1 is involved in cell proliferation, migration and invasion processes, is associated with a poor prognosis and correlates with a glioma grades. The overexpression of LncRNA FOXD3-AS1 reduces the level of transcription factor Forkhead Box D3 (FOXD3), which takes part in the processes of differentiation, proliferation, migration and apoptosis [58].

#### 2.1.8. FTX lncRNA

The newly discovered lncRNA FTX initiates the proliferation process by binding to miR-342-3p, which, in turn, directly binds Astrocyte Elevated Gene-1 (*AEG-1*) [59]. *AEG-1* is an important player in the carcinogenic process in diverse organs and tissues and can act through multiple pathways, including PI3K/Akt, NF-κB, Wnt/β-catenin and MAPK [126]. It makes FTX a very promising target for novel treatments of glioma. However, there is no data about miR-342p expression in glioma tissue [127].

#### 2.1.9. HOTAIR, HOXA11-AS, UCA1 lncRNAs

UCA1, HOTAIR and HOXA11-AS are the most studied up-regulated lncRNAs. UCA1 is involved in the proliferation and migration processes, and its expression positively correlates with overall patient survival. It was shown that UCA1 activates the expression of inhibitor of Apoptosis Stimulating Protein of p5 (*iASPP*) gene by inhibiting the expression of miR-182 [111]. In normal brain tissue, miR-182 regulates apoptosis, proliferation and migration processes by inhibiting the expression of *iASPP* [128]. At the same time, Sun et al. also showed that elevated levels of UCA1 down-regulate miR-122 [109]. In turn, decreased levels of miR-122 are associated with a tumor proliferation, invasion and migration via Wnt/β-catenin signaling pathway [129,130,131]. In addition, inhibition of UCA 1 expression using si-RNA in U87 and U251 cell lines promoted the expression of *cyclin D1* [110]. Several studies showed that lncRNA HOTAIR is a target for miR-326 [69] and miR 148b-3p [66]. The suppressed expression of HOTAIR together with mimics of miR-326 had the strongest inhibitory effect on proliferation, migration and invasion processes in U87 and U251 cell lines. It was shown that a possible target of HOTAIR/miR-326 is *FGF*, which activates the PI3K/AKT signaling pathway [132]. The involvement of HOTAIR and miR-326 in gliomagenesis was also shown in nude mice, where it suppressed tumor growth and prolonged overall survival [69,70]. Besides that the inhibition of HOTAIR expression induces apoptosis, it was also shown that the BET (Bromodomain and Extra-Terminal) family protein BRD4 regulates HOTAIR expression [68]. The decreased expression of lncRNA HOXA11-AS induces a cell cycle arrest and initiates apoptosis in HG44 and U251 cell lines. HOXA11-AS is targeted by miR-140-5p [74], which regulates cell proliferation and invasion via the VEGFA/MMP-2 signaling pathway [133]. The microarray data revealed that there are about 500 genes correlated with a HOXA11-AS3 expression. The suppressed expression of HOXA11-AS3 inhibited the growth of the glioma tumor in nude mouse [75]. Recently, Xu et al. showed that HOXA11-AS decreases the expression of miR-214-3p, which directly targets Enhancer of Zeste 2 polycomb repressive complex 2 subunit (*EZH2*), a component of many signaling pathways, including Wnt/β-catenin, MAPK and Notch [73].

#### 2.1.10. HULC lncRNA

Despite the fact that HULC is one of the most intensively investigated long non-coding RNAs, due to its association with cancer, it’s up-regulation in glioma was shown only in 2016 [76,77]. It was demonstrated that HULC activates proliferation, adhesion and migration processes together with an EMT process through the PI3K/Akt/mTOR signaling pathway [77].

#### 2.1.11. MIR155HG lncRNA

The increase of LncRNA MIR155HG expression initiates an epithelial mesenchymal transition (EMT). It promotes the tumor cell proliferation process in vitro (U87, U251, and primary GBM cells) and in vivo (mouse model system). MIR155HG is a primary RNA for miR-155-3p and miR-155-5p, which targets *protocadherin-9* and -*7*, respectively. Protocadherin-9 and -7, in turn, function as tumor suppressors by inhibiting the Wnt/β-catenin signaling pathway [88,134].

#### 2.1.12. NEAT1 lncRNA

An overexpression of lncRNA NEAT1 in glioma tissue positively correlates with glioma grade [94], and a lower NEAT1 expression correlates with longer survival of glioma patients [92]. NEAT1 activates proliferation, migration and invasion processes and deregulates apoptosis possibly by interacting with miR-449b-5p, which in turn activates the *met* oncogene. The involvement of NEAT1/miR-449-5p/*met* in tumorigenesis was also shown in vivo [94]. In addition, a negative correlation between NEAT1 and miRNA let-7e expression was observed. NEAT1 is a direct target of let-7e and activates cell proliferation via PI3K/AKT/mTOR and MEK/ERK pathways [95]. NEAT1 also activates *met* expression via silencing miR-449b-5p. Met is a well-known oncogene, which activates MAPK/ERK and Akt/mTOR signaling pathways [94]. This is the only study proving the involvement of miR-449b-5p in gliomagenesis. Recently, Yang et al. discovered that NEAT1 increases Cyclin Dependent Kinase 6 (*CDK6*) expression via miR-107 in the U87 cell line [93].

#### 2.1.13. PVT I lncRNA

Yang et al. demonstrated an increased lncRNA PVT I expression in glioma tissue. They showed that PVT I expression positively correlates with poor outcome and alters the expression of *EZH2*. This lncRNA also regulates cell cycle and apoptosis in vitro and in vivo [97,100]. PVT I, by binding to miR-190a-5p and miR-488-3p, is able to regulate the expression of Myocyte Enhancer Factor 2C (*MEF2C*) and *JAGGED1*. It is known that the JAGGED1-NOTCH signaling pathway is critical for glioma proliferation [98].

### 2.2. Down-Regulated lncRNAs in Biological Glioma Processes

The down-regulated lncRNAs function as tumor suppressors and inhibit cell cycle and proliferation. Deregulation of the tumor suppressors is common in all types of cancer.

#### 2.2.1. ADAMTS9-AS2, MDC1-AS, MEG3, TSLC1-AS1 lncRNAs

Little is known about lncRNAs MEG3, MDC1-AS, TSLC1-AS1 and ADAMTS9-AS2. MEG3 activates cell proliferation process through the PTEN pathway via miR-19a [87]. A decreased expression of lncRNA MDC1-AS induces cell proliferation via MDC1, but a detailed mechanism remains to be revealed [86]. TSLC1-AS1 is an antisense regulator of the suppressor gene Tumor Suppressor in Lung Cancer 1 (*TSLC1*). Therefore, the exact pathway by which TSLC1 suppresses cell growth is still unknown [104]. In addition, not much is known about ADAMTS9-AS2, the expression of which is down-regulated in glioma and correlates with a glioma grade. It was shown that an over-expression of ADAMTS9-AS2 inhibits cell migration and invasion processes [41].

#### 2.2.2. CASC2 lncRNA

Liao et al. showed that CASC2 negatively correlates with glioma grade and overall survival in patients. Their data shows that the expression of CASC2 influenced the TMZ resistance through the CASC2/miR181a/PTEN pathway [44]. Wang et al. also showed that CASC2 suppresses Wnt/β-catenin pathway [45]. Another study showed that CASC2 is a target of miR-21 [43], which is frequently overexpressed in tumor cells [135] and takes part in proliferation, migration and invasion processes, possibly through the p53 pathway [136]; but there is no data about the direct involvement of CASC2 in this pathway.

#### 2.2.3. HOTTIP lncRNA

The expression of the lncRNA HOTTIP is mainly down-regulated in glioma. However, its overexpression inhibited proliferation and induced apoptosis in vitro. Also, a negative correlation between HOTTIP and BRE (one of the BRCA1-A complex subunit) levels was observed [72]. BRE, in turn, plays an essential role in preventing replicative and DNA damage-induced premature senescence [137]. The decrease of *BRE* expression, in addition, decreases the expression of *cyclin A* and *CDK2* and increases the expression of *p53* [72]. Interestingly, another study showed that under hypoxic conditions there is an overexpression of HOTTIP RNA, and the HOTTIP expression in glioma samples with metastasis was approximately 4 folds higher than in glioma samples without metastasis. Zhang et al. demonstrated, that HOTTIP contributes to epithelial-mesenchymal transition (EMT) and metastasis in glioma by targeting miR-101 and in turn elevates the amount of *ZEB1* mRNA, which is a direct target of miR-101 [138].

#### 2.2.4. Lnc00462717 lncRNA

Low Lnc00462717 expression negatively correlates with the expression of Mouse double minute 2 (*MDM2*). Bioinformatics analysis revealed that Lnc00462717 is upstream of MDM2, and it directly targets *MDM2* transcription. Lnc00462717 suppresses cell proliferation, survival and migration processes possibly via the MDM2/MAPK pathway, but the specific mechanisms are yet unknown. There is no data about the interactions of Lnc00462717with miRNAs or any other target genes [79].

#### 2.2.5. MALAT1 lncRNA

The controversy regarding the expression pattern of lncRNA MALAT1 in gliomas may be associated with the complexity of non-coding RNA biology and the heterogeneity of glioma tumors. Three studies showed that the expression of MALAT1 is up-regulated in glioma, and two—that the expression is down-regulated. Based on the data, the expression of MALAT1 is down-regulated in glioma tissue and cell lines. Besides, high levels of MALAT1 correlate with better overall survival of patients with glioma. It was also shown that MALAT1 negatively regulates miR-155 expression. One of the validated targets of miR-155 is circular RNA F-box/WD-repeat-containing protein (7FBXW7). According to the results, MALAT1 suppresses the viability of the glioma cells by down-regulating miR-155 and promoting the expression of FBXW7 [84]. Han et al. also measured the expression of MALAT1 in glioma specimens and glioma cell lines. In accordance to their data, the expression of MALAT1 is higher in non-cancerous brain tissues, whereas there were no differences between the grades. An overexpression of MALAT1 is associated with suppressed of glioma cells. This effect was also shown in vivo. Their results revealed that the overexpression of MALAT1 suppressed the activation of the MAPK pathway [83].

On the other hand, referring to Xiang et al.’s results, MALAT1 is highly expressed in glioma tissue and in glioma cell lines [82]. The decrease of MALAT1 expression suppressed the growth rate of glioma cells and induced apoptosis. They also showed that the decreased expression of MALAT1 down-regulated the expression of *cycline D1* and *myc* [82]. Fu et al. also detected the increase of MALAT1 expression. They concluded that MALAT1 is able to induce proliferation and autophagy in the cells and targets miR-101, which, in turn, regulates the expression of Stathmin 1 (*STMN1*), *RAB5A* and Autophagy Related 4D Cysteine Peptidase (*ATG4D*). The increase of miR-101 expression or decrease of MALAT1 expression reduces the expression of autophagy genes [81]. A precise classification of glioma specimens on a molecular level could possibly resolve this controversy.

#### 2.2.6. TUG1 lncRNA

It was shown that the decreased expression of TUG1 correlates with the up-regulation of miR-26a. One of the validated targets of miR-26a is *PTEN* [139]. Using U251 and SHG-44 glioma cell lines, it was shown that TUG1 regulates *PTEN* expression by sponging miR-26a [107]. In another study, it was identified that the up-regulation of TUG1 activates caspases by inhibiting the expression of *Bcl-2*, suggesting its involvement in regulation of programmed cell death and immune response [106].

#### 2.2.7. TUSC7 lncRNA

The expression of TUSC7 is down-regulated in glioma tissue and negatively correlates with the overall survival of the patients and expression of miR-23b [108]. In addition, Jiang et al. showed that the expression of miR-23b regulates the PI3K/Akt signaling pathway [140]. However, there is no evidence of the direct involvement of TUSC7 in PI3K/Akt pathway regulation. Overall, the large list of lncRNAs that are aberrantly expressed in glioma tissue shows its importance for many processes involved in glioma development. These lncRNAs are a potential tool for accurate diagnosis, individualized treatment and prognosis. Furthermore, a deregulated expression of lncRNAs according to chemosensitivity, angiogenesis and brain-tumor-barrier status has been observed and will be discussed in later chapters.

### 2.3. miRNAs Involved in Gliomagenesis in Association with lncRNAs

As miRNAs have been studied longer than lncRNAs, a lot of data has been accumulated about miRNA target genes and signaling pathways. After the discovery of lncRNAs, there are still links missing between processes controlled by miRNA and lncRNA. Not all of these pathways have been linked to associated lncRNAs. Here we summarize the data about signaling pathways and targeted genes which are regulated by miRNAs associated with lncRNAs (Figure 1). According to the target genes, miRNAs and lncRNAs can be grouped into three main signaling pathways: PI3K/Akt/mTOR, Wnt/β catenin and Notch. Some of the miRNAs/lncRNAs are able to regulate more than one signaling pathway. The network of these ncRNAs is more complicated, but it makes them promising prognostic and/or predictive biomarkers (Figure 2).

#### 2.3.1. let-7e

The dual-luciferase assay results showed a potential binding region between NEAT1 lncRNA and let-7e miRNA. An expression of let-7e is significantly down-regulated in GBM tissues and GBM cell lines, compared to healthy brain tissues and normal human astrocytes. Moreover, let-7e expression negatively correlates with glioma grade. A CCK-8 assay demonstrates that glioma stem cell proliferation is significantly lower in the let-7e overexpression group, thus suggesting that let-7e functions as a tumor suppressor. It was showed that *NRAS* may be involved in NEAT1/let-7e-dependent progression of glioma stem cells [95].

#### 2.3.2. miR-19a

LncRNA MEG3 regulates the cell proliferation process through *PTEN* and miR-19a [87]. Sun et al. demonstrated that miR-19 expression positively correlates with glioma tumor grade. Down-regulation of miR-19 expression inhibits proliferation, invasion, induces cell cycle G1 arrest and apoptosis via Runt Related Transcription Factor 3 (*RUNX3*) expression regulation by binding to its 3′-UTR and represses the β-catenin/Transcription Factor 4 (TCF4) transcription activity in LN229 and U87 cells [141]. Bioinformatics analysis showed that Peroxisome proliferator-activated receptor alpha (*PPARα*) is another target of miR-19a, and a low *PPARα* expression level is associated with worse outcomes in clinical glioma patients. PPARα is down-regulated by E2F1/miR-19a signaling in glioma cells. In addition, it was found that miR-19a expression is significantly higher in high grade glioma samples when compared to low grade glioma samples. Down-regulation of miR-19a is associated with suppressed glioma cell proliferation, invasion and aerobic glycolysis [142].

#### 2.3.3. miR-21

Another study showed that CASC2 is a target of miR-21 [43], which is frequently overexpressed in tumor cells [135] and takes part in stimulation of proliferation, migration and invasion processes possibly through p53 pathway [136]. Luo et al. study results show that a high miR-21 expression significantly promotes the migration and invasion of glioma cells through the miR-21/Sox2/β-catenin signaling pathway [143]. Chen el al., by analyzing microarray gene expression data and clinical information of glioma patients, identified seven miRNAs (miR-7, miR-15b, miR-21, miR-124a, miR-129, miR-139, miR-218) with prognostic potential [144]). Moreover, miR-21 is up-regulated in glioma vessels and subsets of glioma cells. The expression of miR-21 is co-localized in vasculature and in tumor cells bordering necrotic areas together with angiogenesis-associated markers HIF-1 and VEGF [145]. Furthermore, evidence shows that miR-21 increases the resistance of human glioma cells to carmustine (BCNU) by decreasing Sprouty RTK Signaling Antagonist 2 (Spry2) protein levels [146].

#### 2.3.4. miR-23b

LncRNA TUSC7 is related to miR-23b. Bioinformatics analysis indicated that TUSC7 specifically binds to miR-23b, which was up-regulated in glioma and negatively correlates with the expression of TUSC7 [108]. Jiang et al. demonstrated that Mitochondrial transcription factor A (*TFAM*), which plays a key role in mitochondrial DNA replication, transcription and inheritance, is the direct target of miR-23b. Moreover, miR-23b expression was significantly lower in glioma when compared to normal tissue and negatively correlated with the glioma malignancy grade. A high miR-23b expression also inhibits cell proliferation, cell cycle progression, migration and colony formation of U251 glioma cells [140].

#### 2.3.5. miR-26a and miR-145

LncRNA TUG1 also regulates the expression of miR-145. Its expression is decreased in glioblastoma tumor tissue and glioma stem cells and negatively correlates with glioma grade. On the other hand, the up-regulation of miR-145 inhibited cell proliferation, invasion, induced G1/S transition arrest in vitro and suppressed xenograft growth in vivo [147]. MiR-145 is one of the miRNAs which directly regulate the expression of Platelet Derived Growth Factor Receptor Beta (*PDGFRB*), neuronal markers, *βIII-tubulin*, neuronal nuclei antigen (*NeuN*), Microtubule Associated Protein 2 (*MAP2*), BCL2 Interacting Protein 3 (*BNIP3*) and the Notch signaling pathway. It regulates microvascularisation, neurogenesis and cell proliferation [148,149,150]. Besides that, it was shown that the expression of miR-26a negatively correlates with TUG1. Studies have shown that miR-26a expression is up-regulated in glioblastoma. MiR-26a also regulates the expression of critical signaling genes such as *PTEN*, *RB1* and *MAP3K2/MEKK2*, promotes tumor growth and inhibits JNK-dependent apoptosis in vivo [151]. All these results show the importance of TUG1 and associated miRNAs in glioma formation and progression processes.

#### 2.3.6. miR-29

An increased expression of miR-29a in primary GBM cells is associated with lncRNAs Xist and H19. miR-29a, by targeting Induced myeloid leukemia cell differentiation protein (*MCL1*), increases apoptosis in GICs cells [152]. Xi et al. demonstrated that miR-29a is down-regulated in glioblastoma stem cells when compared to non-glioblastoma stem cells. A high miR-29a expression in GSCs inhibited cell proliferation, migration, invasion and promoted apoptosis by binding to its direct target Quaking gene isoform 6 (*QKI-6*) [153]. Xiao et al. showed that miR-29c expression was significantly down-regulated in the majority of glioma tumor samples compared to normal brain tissues. Moreover, a high miR-29c expression level was associated with better overall survival in glioma patients and increased sensitivity to temozolomide. miR-29c, by indirectly targeting *MGMT* through Specificity protein 1 (Sp1), inhibited cell growth and promoted apoptosis in U251/TR cell lines in vitro and in vivo [154].

#### 2.3.7. miR-101

It was shown that the expression of miR-101 is associated with two lncRNAs: MALAT1 and HOTTIP. miR-101 epigenetically suppresses the expression of LIM domain only 3 (*LMO3*) by inhibiting the binding of Upstream stimulatory factor 1 (USF) and Myeloid Zinc Finger 1 (MZF1) to the *LMO3* promoter and induces cell apoptosis in U251 glioma cells [155]. An elevated expression of this miRNA inhibits proliferation and migration of glioma cell in vitro, as well as the growth of glioma tumor in vivo via regulation of cyclooxygenase-2 (*COX-2*) and transcription factor *SOX9* expression [156,157]. MiR-101 is also essential for chemotherapy. It was shown that miR-101 expression is associated with sensitivity to temozolomide in TMZ-resistant GBM cells through down-regulation of glycogen synthase kinase 3β (*GSK3β*) expression. In addition, a lower miR-101 expression is related to a worse survival prognosis of GBM patients, thus suggesting its potential role as a prognostic biomarker [158].

#### 2.3.8. miRNA-107

MiRNA-107 is another miRNA which associates with NEAT. The expression of miRNA-107 is down-regulated in glioma cell lines and tissue. An elevated miR-107 expression suppressed growth, migration and invasion abilities of glioma cells, directly targeting *Notch2* and inhibiting *Notch2*, *Tenascin-C*, *MMP-12* and *Cox-2* expression [159]. Chen et al. also showed that a high miR-107 expression inhibits glioma cell proliferation, invasion and down-regulates the expression of *Notch2*, stem cell markers *CD133*, *Nestin* and *MMP-12* in the U87GSC cell line, thus indicating its potential therapeutic role in glioma treatment [160]. A low expression of miR-107 induces apoptosis in glioma cells possibly through the regulation of the FADD/caspase-8/caspase-3/7 signaling pathway [161]. Moreover, a low expression of miR-107 induces glioma angiogenesis through the activation of *VEGF* expression [162]. Furthermore, miR-107 down-regulation is significantly associated with advanced pathological grade in glioma patients, large tumor size and lower Karnofsky performance score. Those patients with a low miR-107 expression had shorter overall survival and progression-free survival rates, compared to patients with elevated miR-107 expression, thus showing its value as a prognostic biomarker in glioma [163].

#### 2.3.9. miR-122

UCA1 also down-regulates the expression of miR-122 [109]. Down-regulation of miR-122 is associated with tumor proliferation, invasion and migration via the Wnt/β-catenin signaling pathway [129,130,131]. Su et al. demonstrated that miR-122 expression is down-regulated in human glioma tissues and glioblastoma stem cells [164]. The dual-luciferase reporter assay revealed that miR-122 directly targets SOX2 Overlapping Transcript (*SOX2OT*) gene and increases its expression. A low expression of *SOX2OT* is associated with inhibited proliferation, migration and invasion of glioblastoma stem cells [164]. Moreover, Yerukala et al. showed that miR-122 expression correlates with the survival time of patients with glioblastoma and could be used as a prediction biomarker [165].

#### 2.3.10. miR-137

It was shown that miR-137 directly regulates the expression of *Rac1*, which plays important role in cell cycle progression, growth and cell motility. Sun et al. also demonstrated that low miR-137 expression in glioma tissue, compared with healthy brain tissue, induces proliferation, inhibited cell cycle arrest and induced cell apoptosis [166]. Li et al. showed that expression of miR-137 in blood serum is down-regulated in GBM patients compared to healthy controls [167]. This down-regulation strongly correlates with glioma clinical grade, Karnofsky performance score and poor survival of GBM patients [167]. MiR-137 also inhibits cell proliferation and angiogenesis of GBM cells by directly regulating the level of Polycomb group protein EZH2 [168]. Thus, an interaction of lncRNA Xist and miR-137 can be a very promising target for the treatment of glioma.

#### 2.3.11. miR-140

Multiple lncRNA interactions with miR-140 also highlight the importance of this miRNA in glioma. H19, MALAT1 and HOXA11-AS down-regulates the expression of miR-140-5p in glioma. Low expression levels of miR-140 correlate with an enhanced cell proliferation, migration and invasion processes. It was shown that miR-140 directly targets *disintegrin* and *metalloproteinase ADAM9* genes [169]. miR-140 is also involved in blood-tumor barrier permeability regulation. Ma et al. showed that nuclear transcription factor Y subunit alpha (*NFYA*) is another target of miR-140 and acts as a transcription factor regulating the expression of *ZO-1*, *occludin* and *claudin-5* [80]. In addition, it was shown that miR-140-5p increases the expression of *VEGFA* and *MMP2* in vitro [133].

#### 2.3.12. miR-142

It has been shown that the expression of tumor suppressive miR-142 is associated with inhibited cancer cell proliferation and invasiveness. LncRNAs may function as miRNA sponges, thus reducing their regulatory impact on mRNAs. Zhang et al. showed an association between lncRNA TP73-AS1 and miR-142, which controls glioma cell growth via the HMGB1/RAGE pathway [103]. Moreover, miR-142 expression is down-regulated in glioma cell lines and tumor tissue and correlates with glioma tumor grade and glioma patients’ survival. Overexpression of miR-142 inhibited glioma cell migration and invasion abilities through regulating *Rac1* and Integrin Subunit Beta 8 (*ITGB8*) expression [170,171]. Rac1 is a GTP-bound protein, involved cell migration and invasion processes and taking part in the PI3K/Akt/mTOR and MAPK signaling pathways [170]. ITGB8 is a member of the integrin family and is involved in tumorigenesis [170].

#### 2.3.13. miR-144

It was shown that TUG1 competes with miR-144. The authors showed that miR-144 can inhibit the expression of *HSF2* which acts as a transcription factor for tight junction proteins. The relationship between TUG1, mir-144, and HSF2 was demonstrated in an animal model system [105]. Lan et al. demonstrated that the level of miR-144-3p was significantly down-regulated in glioma tumor tissue compared with healthy brain tissues and decreased with higher glioma grades [172]. In addition, low miR-144-3p expression was associated with worse overall survival in glioma patients. miR-144-3p inhibited GBM cell proliferation and invasion by suppressing its direct target *met* in vitro and in vivo [172]. A direct correlation with one important oncogene MET makes this miRNA a very promising target.

#### 2.3.14. miR-152

The third known miRNA interacting with Xist is miR-152. The expression of miR-152 is lower in glioma. The invasion assay showed that miR-152 regulates cell invasiveness via binding to its direct target gene *MMP-3* [173]. A low miR-152 expression enhanced cell proliferation, whereas a high miR-152 expression promoted glioma cell apoptosis by regulating the expression of Runt Related Transcription Factor 2 (*Runx2*) [174]. Furthermore, miR-152 directly targets DNA methyltransferase 1 (*DNMT1*) gene. Both miR-152 overexpression and *DNMT1* knockdown are significantly associated with induced apoptosis and inhibited invasiveness [175].

#### 2.3.15. miR-155

On other hand, MALAT1 negatively regulates the expression of miR-155 and positively, the expression of miR-203 and miR-101. The expression of miR-155 is elevated in glioma tumor tissue when compared to the healthy tissue. Contrariwise, a decreased expression of miR-155 inhibited the proliferation of glioma cells and the activation of the Wnt/β-catenin pathway, as well as suppressed the growth of U-87 MG glioma xenografts in mice [176]. Decreasing the level of miR-155 expression sensitizes glioma cells to the temozolomide (TMZ) by targeting the p38 isoforms *MAPK13* and *MAPK14* [177]. The expression of miR-155, in parallel, is regulated by lncRNA MIR155HG. Wu et al. showed that the MIR155HG/miR-155-3p/miR-155-5p axis plays a key role in glioma progression and may be used as a prognostic factor for glioblastoma patient survival [88].

#### 2.3.16. miR-181

Multiple interactions of various lncRNAs with miR-181 indicates the importance of this microRNA in glioma. LncRNA CCAT1 by interacting with miR-181b regulates tumorigenesis and EMT of glioma [48]. Besides, Zhou et al. showed that decreased expression of miR-181b stimulates proliferation, migration and invasion in vitro via negative regulation of *SALL4* (Sal-like protein 4) expression [178]. In addition, the increased expression of miR-181b suppressed the levels of cyclin D1, c-Myc and Ki-67 proteins, essential for cell proliferation [179]. The expression of MiR-181b differs between LGG and HGG glioma tumors, thus indicating that miR-181b expression is associated with glioma malignancy [180]. Evidence shows that miR-181b may be involved in chemosensitivity for its ability to change the sensitivity to teniposide through binding *MDM2* 3′-UTR region [181]. Another miRNA that belongs to the miR-181 family is miR-181d. It is associated with lncRNA NEAT1 and increases BTB permeability [91]. In parallel, it was shown that Insulin-like Growth Factor (*IGF-1*) is a direct target of this miRNA and is relevant in regulating cell growth and cytokine secretions. High IGF-1 level and low miR-181d expression levels were related to poor patient survival, thus suggesting its role as a prognostic factor in glioblastomas [182]. MiR-181d also targets *K-ras*, *bcl-2* [183] and down-regulates *MGMT* expression [184]. The elevated expression of miR-181a is associated with lncRNA CASC2 and modulates the resistance to TMZ [44]. Additionally, Ma et al. showed that up-regulation of miR-181a is also associated with increased BTB and decreased expression of *ZO-1*, *occludin*, and *claudin-5*. Moreover, Kruppel-Like Factor 6 (*KLF-6*) is a direct target of miR-181a and might play an important role in regulating BTB permeability [185].

#### 2.3.17. miR-182

One of UCA1 regulating miRNAs is miR-182, significantly expressed in the oligoneural subclass of GBM. It promotes the apoptotic response of glioma cells to anti-cancer agents (TMZ and RTK inhibitors) by suppressing the expression of *Caspase* and p53 inhibitor *Bcl2L12* genes [186]. Feng et al. also demonstrated that miR-182 regulates the cell cycle and cell migration in vitro, though modifying the expression of Neuritin 1 (*NRN1*) gene—a novel member of the neurotrophic factors family, which has been shown to be associated with tumor malignancy [187]. In addition, Xue et al. showed that miR-182-5p induces the proliferation and invasion of glioma cells by directly targeting *PCDH8* gene (protocadherin 8) [188]. Furthermore, study results demonstrated that levels of circulating miR-182 in glioma patients’ blood were higher in comparison to healthy controls, and the expression of miR-182 is positively associated with Karnofsky performance score and tumor grade. These data suggest that miR-182/UCA1 may be used as novel biomarkers for prognosis in patients with glioma [189].

#### 2.3.18. miR-186, miR-190a-5p, miR-488-3p

Newly identified lncRNA PVT1 is currently known to regulate the expression of 3 miRNAs: miR-186, miR-488-3p and miR-190a-5p. The expression of miR-186 in glioma tissues is significantly lower in comparison to the healthy brain tissue and negatively correlates with increasing pathological glioma grade [99,190]. It is known that miR-186 regulates Caspase3, Cyclin D1, Bcl-2-associated death promoter (BAD) and Microtubule Affinity Regulating Kinase 2 (MARK2) levels by targeting X-Linked Inhibitor of Apoptosis (*XIAP*) and P21 (RAC1) Activated Kinase 5 (*PAK7*) 3′-UTR. A low miR-186 expression was associated with enhanced proliferation, migration, invasion abilities and inhibited apoptosis in glioma stem cells [190]. In addition, this miRNA is also regulated by lncRNA CRNDE [190]. Unfortunately, there is no additional information about the other two other miRNA associated with lncRNA PVT1: miR-488-3p and miR-190a-5p.

#### 2.3.19. miR-200a

Berthois et al. demonstrated for the first time that miR-200a expression is down-regulated in high grade glioma samples compared with low grade glioma [191]. An overexpression of miR-200a promotes sensitivity to temozolomide in GBM cells. Moreover, the DNA repair enzyme O (6)-methylguanine methyltransferase (*MGMT*) was identified as a direct target of miR-200a the suppression of which results in miR-200a expression increment [191]. Despite the evidence of interaction between lncRNA ATB and miR-200a it remains unknown, whether lncRNA ATB expression affects chemosencitivity to TMZ.

#### 2.3.20. miR-203

The expression of miR-203 is regulated by MALAT1 and CCDC26. Snail family transcriptional repressor 2 (SNAI2) plays an important role in epithelial to mesenchymal transition and was identified as a direct target of miR-203. Down-regulation of *SNAI2* expression was associated with EMT inhibition and drug resistance. Lower miR-203 expression was associated with poorer glioblastoma clinical outcomes [192]. Pal et al. showed that miR-203 can regulate cell proliferation and migration in human glioblastoma cells through GAS41/miR-10b axis [193]. On the other hand, it was shown that miR-203 targets Ataxia Telangiectasia Mutated (*ATM*) and the inhibition of *ATM* expression sensitizes the cells to various chemotherapeutic agents [194]. All this data makes MALAT and CCDC26 important targets.

#### 2.3.21. miR-326

Recently, miR-326 was shown to play an important role in glioma progression. Zhou et al. identified NIN1 (RPN12) binding protein 1 homolog (*NOB1*) as a direct target of miR-326, and it is a potential oncogene in human glioma. A high miR-326 expression in A172 and U373 human glioma cell lines caused cell cycle arrest, suppressed cell proliferation and enhanced apoptosis. Moreover, miR-326 may inhibit cell colony formation and reduce growth of a xenograft tumor, thus indicating that miR-326 and *NOB1* are important for glioma progression both in vitro and in vivo [195]. Another miR-326 target is the SMOothened (*SMO*) gene. A high miR-326 expression suppresses the expression of *SMO* and other down-stream genes, and, in turn, inhibits activity of the Hedgehog pathway in U251 tumor stem cells [196]. It was confirmed that HOTAIR lncRNA directly binds miR-326. As *FGF1*, which plays an important role in glioma by activating PI3K/AKT and MEK 1/2 pathways is a target of miR-326, an increased expression of miR-326 reduced *FGF1* expression level [69]. Thus, HOTAIR/miR-326/FGF1 plays an important role in glioma.

#### 2.3.22. miR-342-3p

One of such miRNA is miR-342-3p associated with lncRNA FTX [59]. MiR-342-3p regulates the expression of Astrocyte-elevated gene-1 (*AEG1*). AEG-1 is an important player in the carcinogenic process, in diverse organs and tissues, and it can act through multiple pathways, including PI3K/Akt, NF-κB, Wnt/β-catenin and MAPK [126,197]. In addition, it was shown that miR-342-3p expression was remarkably lower in the GBM group, compared to healthy controls, and could help to separate glioma from healthy controls with high specificity and sensitivity. Also, there were significant differences in miR-342-3p expression levels between glioma patients with grade II, III and IV. Plasma levels of miR-342-3p were particularly decreased in glioma with higher tumor grades [198]. Similar results showed Roth et al. comparing miRNA profiles from the blood in glioblastoma patients and healthy controls [127]. It was determined that miR-342-3p expression was significantly lower in the glioblastoma patient group [127]. It makes lncRNA FTX a very promising target for novel treatment of glioma. However, there is no data about miR-342p expression in glioma tissue [127].

#### 2.3.23. miR-384

Another miRNA which may be associated with glioma progression and lncRNA CRNDE is miR-384. The expression of miR-384 is significantly down-regulated in glioma tissue and glioma cell lines compared to healthy brain tissue and normal human astrocytes. The expression of miR-384 also negatively correlates with glioma progression grade. A high miR-384 expression is associated with inhibited proliferation in U87 and U251 cell lines. It has been shown that miR-384 targets *PIWIL4* 3′-UTR and suppresses glioma cell malignancy via regulating the expression of *cyclin D1*, *VEGFA*, *SNAI2*, *MMP*-9, *Bcl*-2, *Bcl-xL* and *Caspase 3* genes [53].

#### 2.3.24. miR-429

MiR-429, which is a target of Xist, is able to activate endothelial monocyte activating polypeptide-II (EMAP-II). EMAP-II regulates BTB permeability by changing *ZO-1*, *Occludin* and *Claudin-5* expression [199]. However, the role of Xist in BTB via miR-429 was not proved yet. SOX2 is another target of miR-429 [200], and the role of lncRNA in this pathway not revealed yet.

#### 2.3.25. miR-449b-5p

Little is known about the role of miR-449b-5p in glioma. There is only one study showing a negative correlation between miR-449b-5p and lncRNA NEAT expression. Besides, the results revealed that miR-449b-5p directly targets the *met* gene [94].

Increasing numbers of studies have revealed the presence of an interaction network, where ncRNAs could regulate mRNA by binding to its 3′-UTR, thus modifying protein expression [201]. This evidence suggests that lncRNAs are associated with ncRNAs, such as miRNAs, and may play an important role in glioma pathogenesis [202]. The possibility of a single miRNA affecting the expression of hundreds mRNA can help us identify new biological pathways in glioma. The biological role of miRNA in the glioma treatment and tumorigenesis is still in question. Dysregulation of miRNAs expression profiles may be associated with glioma progression by the modulation of cell proliferation, invasion and apoptosis [203]. Therefore, miRNAs may have a clinical potential for a targeted therapy or glioma molecular diagnosis and prognosis [204].

### 2.4. The Potential of Non-Coding RNAs for the Prediction of Resistance to Chemotherapy

Development of drug resistance is the major problem is that observed in glioma treatment. Changes in ncRNA expression are associated with response to treatment and may have value as prognostic and predictive biomarkers. Attempts to identify the lnc-RNAs, associated with chemoresistance, is very important for TMZ-resistant glioma. A few observations support the potential of non-coding RNAs as predictors of sensitivity to TMZ treatment.

#### 2.4.1. H19 lncRNA

It was shown that TMZ resistant patients had higher levels of lncRNA H19. This effect is mediated via major drug resistance genes, such as Multidrug-resistant (*MDR*), Multidrug resistance-associated protein (*MRP*), and ATP Binding Cassette Subfamily G Member 2 (*ABCG2*), the expression of which was altered in both mRNA and protein level [63]. Jia et al. showed that H19 increases chemoresistance by activating EMT via the Wnt/β-catenin pathway [205].

#### 2.4.2. Xist lncRNA, miR-29c

LncRNA Xist is potentially associated with resistance to chemotherapy. Xist is up-regulated in 33.33% of glioblastoma cases and correlates with a glioma grade, tumor size, and survival prognosis. Elevation of Xist expression down-regulates miR-29c, which is also responsible for the chemosensitivity in breast cancer [154,206,207,208]. In addition, it was shown that Xist/miR-29c regulates chemosensitivity via transcription factor gene *SP1* and O-6-Methylguanine-DNA Methyltransferase (*MGMT*) gene expression [116].

#### 2.4.3. miR-181a, miR-193a-5p

Jiang et al. showed that a decreased expression of CASC-2 in glioma is associated with a TMZ-induced autophagy. The study showed that TMZ resistance is determined by miR-193a-5p via the mTOR signaling pathway and inhibition of autophagy restored chemosensitivity [46]. CASC2 also induces chemoresistance by down-regulating the miR-181a/PTEN pathway [44].

#### 2.4.4. MALAT1 lncRNA, miR-203

LncRNA MALAT1 is also observed to influence chemoresistance. It induces TMZ resistance by down-regulating miR-203 which, in turn, increases the expression of thymidylate synthase (*TS*). TS is a key regulator of cell proliferation, and lncRNA MALAT 1 is able to suppress cell proliferation through miR-203-TS pathway in TMZ resistant cell lines [85].

Overall, observed effects to the sensitivity to chemotherapy of glioma cells, suggest the consideration of these specific miRNAs and lncRNAs as potential biomarkers for the therapeutic approach that may lead to a longer progression free and overall survival of glioma patients.

### 2.5. Non-Coding RNAs in Angiogenesis

The major hallmark of cancer is tumor vascularization or sustained angiogenesis [209]. Malignant gliomas are characterized as the most vascular of all solid tumors, therefore angiogenesis is considered to be the main feature of glioma progression. Although the hypoxia is a well-known driver for vascularization, there is also evidence that non-hypoxia driven mechanisms exist [210,211,212]. Moving further, Hardee and Zagzag indicated at least five mechanisms of neovascularization in glioblastomas [213]. The major player of vascularization is growth factor VEGF under reduced oxygen conditions. The hypoxic response in glioblastoma cells is characterized by high levels of hypoxia-inducible factor (HIF-1α) and VEGF [34,210,214]. Transcription factor HIF-1α regulates ~60 genes involved in cell pathways such as anaerobic glycolysis, metabolism, angiogenesis, invasion, and EMT [215]. However, in normoxia, glioma cell angiogenesis is activated through the PI3K/Akt/mTOR pathway. Activation of serine/arginine-rich splicing factor kinase 1 (*SRPK1*) expression which regulates phosphorylation of Akt and controls of VEGF splicing, induces the PI3K/Akt pathway and angiogenesis in glioma under normoxic conditions [216].

#### 2.5.1. H19 lncRNA, miR-29a, miR-675-5p

Currently, a few studies revealed the association between non-coding RNA and angiogenesis. Jia et al. showed, that the expression of lnc-RNA H19 is up-regulated in glioma microvessels compared to normal brain microvessels [65]. The suppressed expression of H19 inhibits tube formation through Vasohibin 2 (VASH2), one of the angiogenesis factors. VASH2 is independent of VEGF and is a new potential target for tumor therapy [217,218,219]. It was shown that H19 also inhibits miR-29a that, in turn, negatively regulates *VASH2* expression [65]. The effect of H19 on angiogenesis was also proved by Jiang et al. in vitro and in vivo [64]. In parallel, it was shown that miR-675-5p, generated from H19, is involved in angiogenesis induced by hypoxia [220].

#### 2.5.2. HULC, POU3F3 lncRNAs

Lang et al. showed that elevated expression of lncRNA-POU3F3 induces angiogenesis through *bFGF*, *VEGFA*, *bFGFR*, and *Angio* genes. LncRNA-POU3F3 can be packed into an exosome and transferred to an endothelial cell where it can induce a formation of new blood vessels [96,221]. Elevation of LncRNA HULC expression induces the expression of Endothelial Cell Specific Molecule 1 (*ESM-1*) gene. ESM-1 is known as a molecular signature of angiogenesis. Its expression is activated by the VEGF/PI3K pathway [77].

#### 2.5.3. Xist lncRNA, miR-137, miR-429

LncRNA Xist was observed to affect angiogenesis. It was shown that reduced expression of Xist reduced angiogenesis in vivo by directly regulating the expression of miR-429 [115]. In parallel, Xist is able to induce angiogenesis via miR-137, which targets tight junction protein gene ZO-2 and Forkhead Box C1 (*FOXC1*), and *FOXC1* is able to promote atypical chemokine receptor 3 (*ACKR3*) expression and activate angiogenesis [113].

#### 2.5.4. HOTAIR lncRNA

A suppressed expression of lncRNA HOTAIR inhibits pro-angiogenetic activities of the glioma cells. It was shown that lncRNA HOTAIR mediates angiogenesis via VEGFA [71]

#### 2.5.5. PVT1, SNHG15 lncRNAs, miR-153, miR-186

An increased expression of lncRNA SNHG15 activates angiogenesis via miR-153, which, in turn, activates the expression of *VEGFA* and *CDC42* genes [222]. High levels of lncRNA PVT1 induce angiogenesis and tube formation by down-regulating miR-186 [99].

Taken together, these studies demonstrate the relationship between an altered lncRNAs expression and the glioma tumor angiogenesis mechanisms. All these lncRNAs are potential targets for the anti-angiogenesis therapies. Further investigations are needed to widen the knowledge in this field.

### 2.6. Non-Coding RNAs and Regulation of Blood-Tumor-Barrier

The blood-tumor barrier (BTB) is one of the factors that influence the efficiency of glioma treatment. This process depends on tight junction proteins, such as cloudin and occludin, and their interaction with multiprotein complex, consisting of cytoplasmic scaffolding ZO proteins [223,224].

Ma et al. showed that suppressed expression of lncRNA MALAT1 increased BTB permeability and decreased the expression of tight junction proteins [80]. Further, it was shown that MALAT1 is a target of miR-140, also regulating BTB permeability and the expression of tight junction proteins. There is a negative correlation between MALAT1 and miR-140. Ma et al. showed that another miR-140 target is the *NFYA* gene that acts as a transcription factor and regulates expression of *ZO-1*, *occludin* and *claudin-5* [80]. Low expression of the lncRNA NEAT1 is associated with miR-181d-5p increment, which in turn regulates BTB permeability and the expression of *ZO-1*, *occludin*, and *claudin-5* through SOX5 [91]. Down-regulation of lncRNA TUG1 expression increased BTB permeability and reduced the expression of *ZO-1*, *occludin* and *claudin-5*. It was shown that lncRNA TUG1 competes with miR-144 for the regulation of tight junction protein expression. Authors showed that miR-144 can inhibit Heat Shock Transcription Factor 2 (*HSF2*) expression. HSF2 acts as a transcription factor for tight junction proteins. The relationship between lncRNA TUG1, mir-144, and HSF2 was also demonstrated in an animal model system [105]. Decreasing the expression of lncRNAs HOTAIR and Xist leads to an increased BTB permeability and a decreased expression of tight junction proteins *ZO-1*, *ZO-2* and *occludin*, together with an increasing expression of miR-148b-3p and miR-137 respectively [67,113]. An up-regulated expression of lncRNA MEG3 increases BTB permeability via the PIWI1/MEG3/mir-330/RUNX3 axis [225].

These results demonstrate that miRNAs and lncRNAs should be investigated more widely, since it might serve as potential targets in regulating the BTB and providing a new strategy for the glioma treatment.

## 3. Discussion

Glioma still remains one of the diseases with worst outcome, despite the progress in diagnostics and treatment. Molecular biomarkers play an essential role in diagnostics and individualized treatment of the patients. According to the new World Health Organization (WHO) classification of gliomas, the application of protein and/or nuclear acid based molecular tests are necessary for integrated diagnosis of gliomas. These routine laboratory tests involve the identification of isocitrate dehydrogenase (*IDH1/2*) mutation, *1p/19q* co-deletion, H3 Histone Family Member 3A (*H3F3A*) mutation, gene for Histone (H3*HIST1H3B*/C) mutation, *RELA* fusion, Telomerase Reverse Transcriptase (*TERT*) promoter mutation, *KIAA1549/BRAF* fusion and *BRAF-V600E* mutation [226,227]. However, evidence shows that these mutations also could be regulated by miRNAs. Cheng et al showed that in total 361 of 487 miRNAs were differentially expressed, depending on the *IDH1*/*2* mutation. Four miRNAs (miR-10b, miR-130b, miR-1304 and miR-302b) were highlighted as a marker for a high or low-risk of poor prognosis associated with the *IDH1*/*2* mutations in lower-grade glioma patients [228].

Microarray data analysis revealed that the expression of 13 lncRNAs is associated with the *IDH1* mutation status. KIAA0495 and HOTAIRM1 were the most down-regulated, and LOC254559 together with LINC00689 were most up-regulated lncRNAs [229]. However, there is still no data about the roles of these RNAs in glioma formation. Furthermore, Jha et al. showed that *H3F3A* mutation is associated with up-regulation of miR-17/92 and its paralog clusters (miR106b/25 and miR-106a/363) and down-regulation of miRNAs belonging to miR379/656 cluster in pediatric high-grade gliomas. In total, 97 miRNAs (62 up- and 35 down-regulated) and 74 miRNAs (21 up- and 53 down-regulated) were found to be *H3F3A* mutation and wild-type specific, respectively. The expression of four miRNAs (miR-15a, miR-424, miR-30e, and miR-378c) was higher in samples with *H3F3A* mutation, when compared to wild type [230]. These findings suggest that miRNAs can be used as specific biomarkers for identification of specific mutations and play an important role in mutation specific gene profiles. Further associations of these miRNAs with treatment outcomes would facilitate the development of novel tools for treatment individualization.

In addition, other less investigated non-coding RNA species, including small nucleolar RNAs (snoRNAs, scaRNA), piwi RNAs (piRNAs) and circular RNAs (circRNAs), have a potential for application as future biomarkers of glioma. Small nucleolar RNAs (snoRNAs) play roles in many biological functions, including biochemical modifications of other RNAs, precursors of miRNA, splicing, and telomerase activity [231]. The genome wide analysis of snoRNA in pediatric gliomas revealed that there were 118 differentially up-regulated (three scaRNAs, 17 snoRNAs, 26 HACA-Box and 72 CD-Box) and 39 down-regulated (38 CD-Box and one snoRNA) small-noncoding RNAs [230]. It was shown that snoRNA SNORD regulates EMT and sensitivity to temozolomide [232]. C/D box snoRNA U76 (SNORD76) inversely associates with an expression of HOTAIR and affects cell proliferation in vitro and in vivo. The elevated expression of SNORD76 reduces the expression of *cyclin A1*, *cyclin B1* and *p107* genes and an increase the expression of *Rb* gene [233]. Up-regulation of snoRNA SNHG18 increases glioma cell radioresistance by repressing Semaphorin 5A [234]. Currently, there is no data about SNHG18 expression pattern in glioma tissue. Recently, the overexpression of snoRNA SNHG1 in glioma tissue was determined, and it correlated with increased tumor cell proliferation and reduced apoptosis; however, the exact cellular pathway and biological targets remain to be discovered [235]. The biological function of scaRNA in glioma is still unknown, but there are several reports where their expression varied in other types of cancer [236,237]. The circular RNA is another group of non-coding RNA serving as a miRNA sponges and sharing the same binding sites [238,239]. Recently, some circRNAs were identified as significant players in glioma development. It was shown that circ-TTBK2 is up-regulated in glioma and cell lines, such as U87 and U251. The down-regulation of circ-TTBK2 together with the overexpressed miR-217 inhibited tumor growth in mice [240]. Furthermore, the expression of circ-FBXW7 is down-regulated in glioma compared to normal brain tissue and higher cicr-FBXW7 expression levels are associated with a better overall survival of glioma patients [241]. Circular RNA CircZNF292 is involved angiogenesis, besides, it’s overexpression in U87MG and U251 glioma cell lines and activates cell proliferation processes via the Wnt/β-catenin signaling pathway. Unfortunately, there is no data about circZNF292 expression in glioma tissue yet [242]. piRNAs, another type of small non-coding RNAs, is generated from precursor RNA which differs from miRNA precursors [243]. piRNA regulates gene expression by triggering epigenetic modification [244] and induces mRNA degradation [245]. PIWI/piRNA complexes regulate cell proliferation, differentiation and cell death [246,247,248]. In addition, piRNA can interact with miRNA and siRNA and modulate cellular processes [249,250]. BTB regulation via the PIWI1/MEG3/miR-330/RUNX3 axis illustrates the complexity and importance of all ncRNA in biological processes [225].

## 4. Conclusions

Brain tumors, especially glioblastoma, still remain one of the most lethal tumors despite of the modern and improved surgical and radiotherapy treatment. More efficient treatment options have to be developed in the future. Non-coding RNA could be the promising tool for targeted therapy combined with other new promising therapies. The use of a complete network of all non-coding RNAs involved in glioma formation and progression could supplement other therapeutic opportunities such as immunotherapy and gene therapy and could also evolve as stratification tool for treatment individualization resulting in a better antitumor benefit.

## Figures and Tables

**Figure 1 cancers-11-00017-f001:**
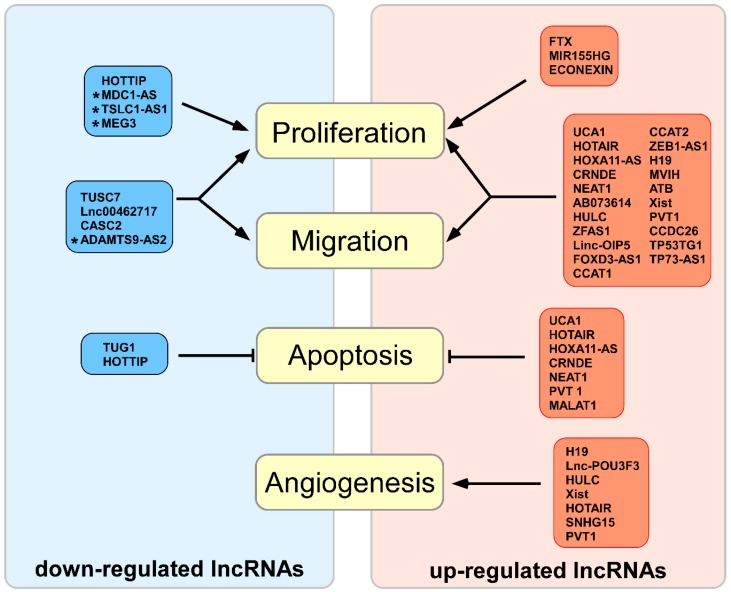
Up-regulated and down-regulated long non-coding RNAs in glioma and their influence on the biological processes. Red indicates up-regulated RNAs in glioma, and blue indicates down-regulated RNAs in glioma. Arrows indicate increased activity of the process; hammerhead arrows indicate decreased activity of the process; asterisk indicates poorly characterized lncRNAs.

**Figure 2 cancers-11-00017-f002:**
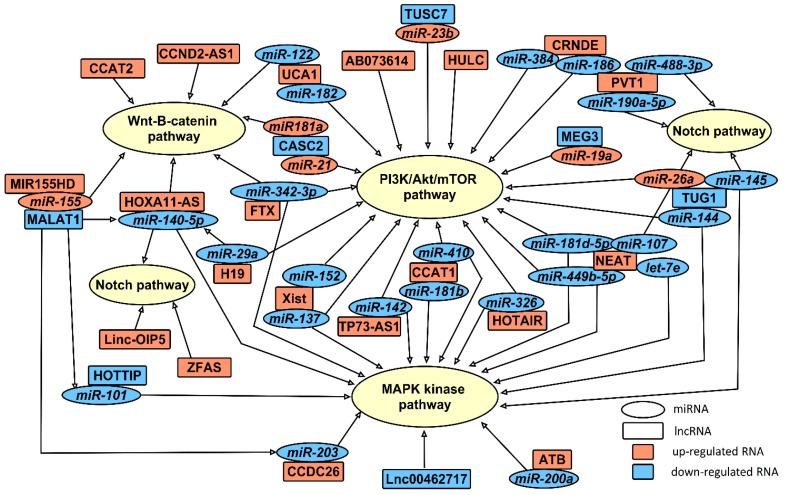
LncRNA and miRNA pathway regulation network. Rectangles represent long-non-coding RNA and ovals-miRNA which are the target/targets of lncRNA. Red color indicates up-regulates RNA in glioma, and blue-down-regulated RNA in glioma. Arrows indicate the signal transduction pathway affected.

**Table 1 cancers-11-00017-t001:** Long non-coding RNAs altered in glioma.

lncRNA	Target miRNA	Genes and/or Pathways	Samples	Biological Processes	References
AB073614 ↑	-	E-cadherin, Vimentin	80 glioma	Proliferation, invasion, migration	[38]
	-	PI3K/AKT	GG-13, HGG-15	Proliferation, invasion, migration, apoptosis	[39]
	-	-	65 glioma	Poor survival	[40]
ADAMTS9-AS2 ↓	-	ADAMTS9, DNMT1	LGG-46, HGG-24	Proliferation, migration, invasion, correlates with survival	[41]
ATB ↑	miR-200a	TGF-β2	79 glioma	Proliferation, migration, invasion	[42]
CASC2 ↓	miR-21	-	LGG-12, HGG-12	Proliferation, migration, invasion, apoptosis	[43]
	miR-181a	PTEN pathway	LGG-30, HGG-27	Proliferation, chemoresistance to TMZ	[44]
	-	β-catenin, cyclin D1 and c-Myc	LGG-26, HGG-21	Proliferation	[45]
	miR-193a-5p	mTOR	LGG-15, HGG-17	TMZ sescitivity	[46]
CCAT1 ↑	miR-181b	FGFR3, PDGFRA	LGG 45, HGG 41	Proliferation	[47]
	miR-410	-	28 glioma	Proliferation	[48]
CCAT2 ↑	-	Wnt/β-catenin signal pathway	LGG-58, HGG-76	Proliferation, cell cycle progression, migration	[49]
CCDC26 ↑	miR-203	-	40 glioma	Proliferation, migration,	[50]
CCND2-AS1 ↑	-	Wnt/β-catenin signaling	54 glioma	Proliferation	[51]
CRNDE ↑	-	-	LGG-46, HGG-118	Proliferation	[52]
	miR-384	PIWIL4	LGG-15, HGG-15	Proliferation, invasion, migration, apoptosis	[53]
	-	mTOR	37 glioma	Proliferation, invasion, migration	[54]
	-	-	LGG-5, HGG-14	Proliferation, apoptosis	[55]
ECONEXIN ↑	miR-411-5p	TOP2A	40 glioma	Proliferation	[56]
FER1L4 ↑	-	-	LGG-335, HGG-21	Proliferation, invasion, apoptosis	[57]
FOXD3-AS1 ↑	-	FOXD3	LGG-13, HGG-31	Proliferation, invasion, migration	[58]
FTX ↑	miR-342-3p	AEG1	LGG-76, HGG-81 from data base; LGG-11, HGG-11	Proliferation, invasion	[59]
H19 ↑	miR-140	iASPP	28 glioma	Proliferation	[60]
	miR-675	-	LGG-15, HGG-20	Proliferation	[61]
	miR-675	CDK6	LGG-4, HGG-4	Proliferation	[62]
	-	MDR, MRP, ABCG2	TMZ-Sensitive-44, TMZ-Resistant-25	TMZ-Resistance	[63]
	miR-29a	-	30 glioma	Angiogenesis	[64]
	miR-29a	VASH2	LGG-5, HGG-5	Angiogenesis	[65]
HOTAIR ↑	miR-148b-3p	-	26 astrocytoma (7 grade II, 19 grade III), 50 oligodendroglioma (38 grade II, 12 grade III) and 81 GBM	Cell growth, proliferation, invasion	[66]
	miR-148b-3p	ZO-1, OCLN, CLDN5, USF1	-	Angiogenesis	[67]
	-	BRD4	17 glioma	Proliferation, invasion, migration, apoptosis	[68]
	miR-326	FGF1, PI3K/AKT and MEK1/2 signal pathways	LGG-6, HGG-6	Proliferation, invasion, migration	[69]
	-	-	67 glioma	Proliferation, invasion, migration	[70]
	-	VEGFA	-	Angiogenesis	[71]
HOTTIP ↓	-	BRE, cycA and CDK2, p53	LGG-37, HGG-48	Proliferation, apoptosis	[72]
HOXA11-AS ↑	miR-214-3p	EZH2	45 glioma	Proliferation, migration, invasion	[73]
	miR-140-5p	-	LGG-13, HGG-30	Proliferation, invasion, migration, apoptosis	[74]
HOXA11-AS3 ↑	-	-	LGG-24, HGG-23	Proliferation	[75]
HULC ↑	-	-	LGG-10, HGG-60	Proliferation, patients overall survival	[76]
	-	Survivin, c-Myc, Cyclin A/D1/E, p-Rb, Skp-1/2, CDK2/4 and EZH2, Bcl-2/Bax, caspase-3/8	LGG-30, HGG-90	Proliferation invasion, migration, angiogenesis, adhesion	[77]
Linc-OIP5 ↑	-	Yap, Notch	LGG-69, HGG-98	Proliferation, migration	[78]
lnc00462717 ↓	-	MDM2	LGG-16, HGG-64	Proliferation, migration, apoptosis	[79]
MALAT1 ↑	miR-140	Nuclear factor YA	LGG-8, HGG-8	Proliferation, chemosensitivity, BTB permeability	[80]
	miR-101	STMN1, RAB5A, ATG4D	32 glioma	Proliferation, migration, autophagy	[81]
	-	CCND1, MYC	LGG-37, HGG-37	Proliferation, apoptosis	[82]
MALAT1 ↓	-	Ki-67, MMP2, MAPK	LGG-32, HGG-100	Proliferation	[83]
	miR-155	FBXW7	LGG-21, HGG-45	Proliferation, correlates with survival	[84]
	miR-203	TS	180 glioma	Chemoresistance	[85]
MDC1-AS ↓	-	MDC1, CycB1/CDK2	15 glioma	Proliferation	[86]
MEG3 ↓	miR-19a	PTEN	40 glioma	Proliferation, apoptosis	[87]
MIR155HG ↑	miR-155-5p, miR-155-3p	PCDH7, PCDH9, Wnt/B-catenin	225 glioma	Proliferation	[88]
miR210HG ↑	-	-	28 glioma	Proliferation	[89]
MVIH ↑	-	-	LGG-57, HGG-70	Proliferation, invasion, migration	[90]
NEAT1 ↑	miR-181d-5p	SOX5	-	BTB	[91]
	-	-	LGG-23, HGG-71	Proliferation, invasion, migration	[92]
	miR-107	CDK6	LGG-5, HGG-24	Proliferation, invasion, migration	[93]
	miR-449b-5p	c-met	LGG-10, HGG-5	Proliferation, invasion, migration, apoptosis	[94]
	let-7e	PI3K/AKT/mTOR, MEK/ERK pathways	LGG-60, HGG-60	Proliferation	[95]
POU3F3 ↑	-	-	LGG-37, HGG-45	Proliferation, angiogenesis	[96]
PVT1 ↑	-	-	98 glioma	Patients survival, chemotherapy, radiotherapy response	[97]
	miR-488-3p	MEF2C	LGG-9, HGG-10	Proliferation, migration, invasion	[98]
	miR-186	Atg7, Beclin1	-	Proliferation, migration, angiogenesis	[99]
	miR-190a-5p	EZH2, JAGGED1	LGG-40, HGG-40	Cell cycle, apoptosis	[100]
SPRY4-IT1 ↑	-	-	LGG-73, HGG-90	Proliferation, invasion, migration, apoptosis	[101]
TP53TG1 ↑	-	-	LGG-12, HGG-12	Proliferation, invasion, migration, apoptosis	[102]
TP73-AS1 ↑	miR-142	HMGB1	LGG-26, HGG-21	Proliferation, invasion	[103]
TSLC1-AS1 ↓	-	TSLC1	LGG-37, HGG-28	Proliferation, migration, invasion	[104]
TUG1 ↑	miR-144	HSF2, ZO-1, OCLN, CLDN5	LGG-5, HGG-5	Angiogenesis	[105]
TUG1 ↓	-	BCL-2, CASP3, CASP9	LGG-57, HGG-63	Proliferation, apoptosis	[106]
	miR-26a	PTEN	LGG-9, HGG-11	Proliferation, apoptosis	[107]
TUSC7 ↓	miR-23b	TUSC7	LGG-19, HGG-20	Proliferation, apoptosis, correlates with survival	[108]
UCA1 ↑	miR-122	-	63 glioma	Proliferation, migration, invasion, apoptosis	[109]
	-	cycD1	LGG-22, HGG-42	Proliferation	[110]
	miR-182	iASPP	LGG-27, HGG-49	Proliferation, migration, invasion, apoptosis	[111]
Xist ↑	miR-137	Rac1	LGG-9, HGG-21	Proliferation	[112]
	miR-137	XCR7, ZO-2, FOXC1	-	Angiogenesis	[113]
	miR-152	-	Astrocytoma	Proliferation, migration, invasion, apoptosis	[114]
	miR-429	-	157 glioma	Proliferation, migration, invasion, angiogenesis	[115]
	miR-29c	MGMT, SP1	LGG-33, HGG-36	Chemosensitivity	[116]
ZEB1-AS1 ↑	-	CycD1 and CDK2 ZEB1, MMP2, MMP9, CDH2, Integrin-β1	LGG-37, HGG-45	Proliferation, invasion, migration, apoptosis	[117]
ZFAS1 ↑	-	Notch signaling pathway	46 glioma	Proliferation, migration, invasion, disease prognosis	[118]

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
