# Peer review of "Non-Coding RNAs in Glioma"

_cancers, 2018, doi:10.3390/cancers11010017_

Reviewer 1 Report

 This excellent and very comprehensive review gives a good overview on the complicated topic of non-coding RNA in glioblastoma. All relevant data is discussed and cited. The conclusions drawn are comprehensible. The linguistic style is rather good, allowing publication without extensive changes necessary. The topic of this review is highly up to date and therefore will find many interested readers. In summary, I strongly support publication without further changes needed.

Author Response

Dear Sir/Madam: Thank you very much for reviewing our manuscript. We appreciate you time devoted to evaluate the manuscript and are grateful for your support.

Reviewer 2 Report

The review on “Noncoding RNAs in glioma” by Rynkeviciene et al contains a rather impressive but not well-organized amount of information. As such it may discourage readers and lose of interest. To avoid this I would suggest some reshaping as follows:

Chapters enumerating lncRNA (2.1 and 2.2) should be re-organized having the name of the lncRNA in bold at the beginning of each paragraph as a mini-title.

Same for miRNA in part 3: the name of the miRNA can be added in bold at the beginning of the paragraph.

Other points.

Sub-title of part 2.1 should read “Up-” and not “Down-” regulated lncRNA.

As little is known about several lncRNAs (see lines 295-304) these same lncRNAs should be distinguished from others in figure 1 (asterisk, question mark, italics…)

Title of chapter 3 should probably emphasize miRNAs, that look like the major players in this phase.

References in lines 397-426 should be formatted according to the journal style.

What is the least aggressive subclass of GBM (lines 442-3)?

In chapter 4 different mechanisms are quoted with relationship to the control of sensitivity to chemotherapy: authors should stress that while this is an important field of research data are not converging and appear less than mature.

The sentence in lines 664-5 should be completed.

Conclusions (chapter 8) are quite generic and could be removed.

The English need to be re-checked.

Author Response

Response to Reviewer 1 Comments

Point 1: The review on “Noncoding RNAs in glioma” by Rynkeviciene et al contains a rather impressive but not well-organized amount of information. As such it may discourage readers and lose of interest. To avoid this I would suggest some reshaping as follows:

Chapters enumerating lncRNA (2.1 and 2.2) should be re-organized having the name of the lncRNA in bold at the beginning of each paragraph as a mini-title.

Same for miRNA in part 3: the name of the miRNA can be added in bold at the beginning of the paragraph.

Response 1    We’ve rearranged the manuscript adding bolded mini title not just for chapters 2,1, 2,2 and 3, but for the entire manuscript.

Point 2:  Other points.

Sub-title of part 2.1 should read “Up-” and not “Down-” regulated lncRNA.

Response 2: We appreciate the note, we’ve corrected this error.

Point 3:  As little is known about several lncRNAs (see lines 295-304) these same lncRNAs should be distinguished from others in figure 1 (asterisk, question mark, italics…)

Response 3: We have changed the figure 1 by labeling indicated lncRNAs by asterisks and explained the meaning of asterisk in the legend of figure 1.

Point 4:  Title of chapter 3 should probably emphasize miRNAs, that look like the major players in this phase.

Response4: We’ve changed the title to: 3. miRNA important for gliomagenesis in association with lncRNA.

Point 5: References in lines 397-426 should be formatted according to the journal style.

Response 5: We’ve corrected indicated references.

Point 6:  What is the least aggressive subclass of GBM (lines 442-3)?

Response 6: based on published information we consider oligoneural subclass of GBM as least aggressive subclass.

Point 7:  In chapter 4 different mechanisms are quoted with relationship to the control of sensitivity to chemotherapy: authors should stress that while this is an important field of research data are not converging and appear less than mature.

Response 7: We have rephrased the text of the chapter 4 to indicate that non-coding RNAs may have the potential as predictors of sensitivity of tumors to chemotherapy and “Few observations support the potential of non-coding RNAs as predictors of sensitivity to TMZ treatment”.

Point 8: The sentence in lines 664-5 should be completed.

Response 8: We have corrected the editing error. The full sentence now reads: HSF2 acts as a transcription factor for tight junction proteins.

Point 9:  Conclusions (chapter 8) are quite generic and could be removed.

Response 9: We’ve deleted the previous conclusions and re-wrote it according to other reviewer suggestions to stress the potential of non-coding RNAs as biomarkers and potential targets for individual treatment options.

Point 10:  The English need to be re-checked.

Response 10: The present version of the manuscript has been edited by the native English speaker.

Reviewer 3 Report

The review of Rynkeviciene et al present a literature review of the role of non-coding in gliomas.

These are my comments:

The manuscript should be fully edited: I would recommend having a native English speaker edit the text.

The introduction should be re-written: it is too superficial about the two topics, gliomas and non-coding RNAs. I suggest let the introduction only for gliomas and explain more in detail about the classification and % of patients in each glioma type; and the clinical and molecular features of the major glioma types. It should be emphasized that GBM is the most common and malignant of all glioma types: see recent reviews (2018) for the WHO classification of glioblastoma 2016.

With the new technologies and instrumentation the major problem with glioma is not the diagnosis but the treatment. Therefore, these point should be clarified in the introduction: maybe as a targets for therapy or predictors of drug/radio therapy.

The second part of the introduction could be a new section about the molecular-biology of nc-RNAs. It should be better explain about the regulatory role of lnc-RNAs at all levels: DNA, RNA and proteins. In addition, it should be mentioned that, through the evolution, lnc-RNA are less conserved that miRNAs.

It is important to mention that for primary brain tumors, the term metastatic is not “correct”. These cells are infiltrating in nature. In addition GBM is currently called “glioblastoma” alone; not “glioblastoma multiforme”.

For all other sections of the review, when lnc-RNAs and miRNA described, it should be clarify where the finding swhere reported: the type of glioma (GBM, etc): this point is important as the expression of lnc-RNA and miRNA are different for each tumor type. This clarification should be performed also in Table 1. An addional column of the type of glioma where the findings were reported.

The Figure 2 must be enlarged.

Line 225: not clear.

Line

Line 225: not clear.

Line 440: not clear

Line 580: the subtitle is not clear

Line 581: not clear

The authors mention the BTB, but how about the BBB?

Lines 682-683 not clear

Conclusions: should be focus on gliomas

In the introduction the authors mention the potential use of nc-RNA for glioma diagnosis however, this point of view was not mentioned in the discussion or in the conclusion.

Author Response

Response to Reviewer 1 Comments

Point 1: The manuscript should be fully edited: I would recommend having a native English speaker edit the text.

Response 1: The present version of the manuscript has been edited by the native English speaker.

Point 2: The introduction should be re-written: it is too superficial about the two topics, gliomas and non-coding RNAs. I suggest let the introduction only for gliomas and explain more in detail about the classification and % of patients in each glioma type; and the clinical and molecular features of the major glioma types. It should be emphasized that GBM is the most common and malignant of all glioma types: see recent reviews (2018) for the WHO classification of glioblastoma 2016.

With the new technologies and instrumentation, the major problem with glioma is not the diagnosis but the treatment. Therefore, these point should be clarified in the introduction: maybe as a targets for therapy or predictors of drug/radio therapy.

The second part of the introduction could be a new section about the molecular-biology of nc-RNAs. It should be better explain about the regulatory role of lnc-RNAs at all levels: DNA, RNA and proteins. In addition, it should be mentioned that, through the evolution, lnc-RNA are less conserved that miRNAs.

Response 2: We re-wrote the introduction according to Your suggestions.

Point 3: It is important to mention that for primary brain tumors, the term metastatic is not “correct”. These cells are infiltrating in nature. In addition GBM is currently called “glioblastoma” alone; not “glioblastoma multiforme”.

Response 3: we’ve changed glioblastoma multiforme to glioblastoma

Point 4: For all other sections of the review, when lnc-RNAs and miRNA described, it should be clarify where the finding swhere reported: the type of glioma (GBM, etc): this point is important as the expression of lnc-RNA and miRNA are different for each tumor type. This clarification should be performed also in Table 1. An addional column of the type of glioma where the findings were reported.

Response 4: there were only a few papers where the specimens were characterized. The most authors merged samples to low-grade and high grade or even just analyzed all glioma samples vs normal tissue without any detailed information. We have replenished missing data as much as possible.

Point 5: The Figure 2 must be enlarged.

Response 5: we have enlarged the figure 2.

Point 6: Line 225: not clear; Line 225: not clear; Line 440: not clear; Line 580: the subtitle is not clear; Line 581: not clear; Lines 682-683 not clear

Response 6: we’ve re-wrote all these unclear sentences and subtitle.

Point 7: The authors mention the BTB, but how about the BBB?

Response 7: based on the published information we consider that current knowledge of non-coding RNAs influence for BBB is too fragmented and therefore we did not include such chapter in this review manuscript.

Point 8: Conclusions: should be focus on gliomas

Response 8: Conclusions has been re-written according suggestion.

Point 9: In the introduction the authors mention the potential use of nc-RNA for glioma diagnosis however, this point of view was not mentioned in the discussion or in the conclusion.

Response 9: We have changed discussion and mainly the conclusion to stress the potential of non-coding RNAs as biomarkers and potential targets for individual treatment options.

Reviewer 4 Report

In their manuscript Rynkeviciene et al., present a very thorough overview on ncRNAs involved in Glioma. Their manuscript highlights the role of both long and short lncRNAs, through several examples of mechanistically elucidated RNA species along with their involvement in several key processes of the disease (e.g angiogenesis, blood-brain barrier, chemosensitivity etc). Therefore this manuscript can address an experienced audience in the field of glioma ncRNAs that wants to be updated and in parallel a less experienced reader that want to be introduced. I refer below to a few minor spelling mistakes that the authors need to correct prior to publication.   

Throughout the text please convert lncRNA to lncRNAs when referring to plural (e.g lines 90-97, 108 etc). Also please improve quality of both figures.

Also please correct the corresponding parts below :

Line 73 : Change “One of promising classes of potential biomarkers…” to “One promising class of potential biomarkers…”

Line 75 : Change “The whole genome gene expression analysis …” to “Whole transcriptome analysis…”

Line 87 : Change “LncRNA usually similar …” to “LncRNA are usually similar…”

Lines 87-88 : Change “regulated by multiple type of  transcription factors“ to “regulated by multiple types of  transcription factors”

Lines 87-88  : Change “generally are spiced and has a poliA tail “ to “generally are spliced and have a poly-A tail”

Line 108 : Change “An aberrant …” to “Aberrant…”

Line 116  : Change “directly interacts …” to “directly interact…”

Line 280  : Change “group of tumor suppressors that role is to inhibit…” to “group of tumor suppressors that inhibit...”

Line 287  : Change “AS1 is another long non-coding RNA, serving as a tumor suppressor and down-regulated in glioma …” to “AS1 is another tumor suppressing long non-coding RNA,  which is down-regulated in glioma...”

Line 329-330  : Change “MALAT1 suppressed  the proliferation of the glioma cells…” to “MALAT1 suppressed  proliferation of glioma cells…”

Line 413  : Change “On other hand, an elevated miR-145 …” to “On the other hand, elevated miR-145 …”

Line 490  : Change “Another studies …” to “Other studies …”

Line 567  : Change “there is no direct evidences …” to “there is no direct evidence …”

Line 660  : Change “and that the main feature …” to “,serving as the main feature …”

Line 669  : Change “Chang et al. study …” to “Chang et al. …”

Line 672  : Change “which involvement …” to “with involvement …”

Line 714  : Change “A decreasing the expression of lncRNA NEAT1 …” to “Downregulation of the lncRNA NEAT1 …”

Author Response

Point 1: Throughout the text please convert lncRNA to lncRNAs when referring to plural (e.g lines 90-97, 108 etc). Also please improve quality of both figures.

Response 1: We corrected the plurality of lncRNA and miRNA and made the bigger text in the figures.

Point 2: Line 73 : Change “One of promising classes of potential biomarkers…” to “One promising class of potential biomarkers…”

Line 75 : Change “The whole genome gene expression analysis …” to “Whole transcriptome analysis…”

Line 87 : Change “LncRNA usually similar …” to “LncRNA are usually similar…”

Lines 87-88 : Change “regulated by multiple type of  transcription factors“ to “regulated by multiple types of  transcription factors”

Lines 87-88  : Change “generally are spiced and has a poliA tail “ to “generally are spliced and have a poly-A tail”

Line 108 : Change “An aberrant …” to “Aberrant…”

Line 116  : Change “directly interacts …” to “directly interact…”

Line 280  : Change “group of tumor suppressors that role is to inhibit…” to “group of tumor suppressors that inhibit...”

Line 287  : Change “AS1 is another long non-coding RNA, serving as a tumor suppressor and down-regulated in glioma …” to “AS1 is another tumor suppressing long non-coding RNA,  which is down-regulated in glioma...”

Line 329-330  : Change “MALAT1 suppressed  the proliferation of the glioma cells…” to “MALAT1 suppressed  proliferation of glioma cells…”

Line 413  : Change “On other hand, an elevated miR-145 …” to “On the other hand, elevated miR-145 …”

Line 490  : Change “Another studies …” to “Other studies …”

Line 567  : Change “there is no direct evidences …” to “there is no direct evidence …”

Line 660  : Change “and that the main feature …” to “,serving as the main feature …”

Line 669  : Change “Chang et al. study …” to “Chang et al. …”

Line 672  : Change “which involvement …” to “with involvement …”

Line 714  : Change “A decreasing the expression of lncRNA NEAT1 …” to “Downregulation of the lncRNA NEAT1 …”

Response 2: We changed all corrected parts.

Round  2

Reviewer 3 Report

Most of the recommendations rised during the first revision were made in this new version.

Double-check references 

Author Response

Point 1: Double-check references 

Response 1: References were re-checked using Endnote software and manually by randomly selecting approx. 10 references.